# Design and Analysis of Cancer Clinical Trials for Personalized Medicine

**DOI:** 10.3390/jpm11050376

**Published:** 2021-05-04

**Authors:** Sin-Ho Jung

**Affiliations:** Department of Biostatistics and Bioinformatics, Duke University, Durham, NC 27710, USA; sinho.jung@duke.edu

**Keywords:** enrichment trial, interaction, predictive biomarker, prognostic biomarker, progression-free survival, stratified randomization trial

## Abstract

Biomarkers play a key role in the development of personalized medicine. Cancer clinical trials with biomarker should be appropriately designed and analyzed reflecting the various factors, such as the phase of trials, the type of biomarker, the study objectives, and whether the used biomarker is already validated or not. In this paper, we demonstrate design and analysis of two phase II cancer clinical trials, one with a predictive biomarker and the other with a prognostic biomarker. A statistical testing method and its sample size calculation method are presented for each of the trials. We assume that the primary endpoint of these trials is a time to event variable, but this concept can be used for any type of endpoint with associated testing methods. The test statistics and their sample size formulas are derived using the large sample approximation based on the martingale central limit theorem. Using simulations, we find that the test statistics control the type I error rate accurately and the sample sizes calculated using the formulas maintain the statistical power specified at the design stage.

## 1. Introduction

In many cancer clinical trials, different types of biomarkers are measured from the tumor, blood or urine using molecular, biochemical, physiological, anatomical, or imaging method at the baseline or during treatment. The observed biomarkers are used for various purposes during the diagnosis and treatment of the diseases. For example, cancer biomarkers are used to diagnose diseases (diagnostic biomarker), to predict the response to a specific treatment (predictive biomarker), to measure the aggressiveness of a disease for patients with no or a non-targeted treatment(prognostic biomarker), to monitor the recurrence of a disease, and so on.

These biomarkers can be used to select a treatment of cancer patients. However, biomarkers should be validated before being used to select a treatment in clinical trials. If a biomarker has not been validated yet, it can be used as a stratification factor of a randomized clinical trial. In such a trial, the biomarker is used for its validation, rather than for treatment selection.

The design and analysis method of a clinical trial with a biomarker-guided treatment can be very different depending on the type of the used biomarker, the biomarker’s development stage, the study objective, and so on. Various design issues of randomized clinical trials with biomarkers have been widely discussed [1]. A series of statistical testing has been proposed for a randomized phase II trial with a potentially predictive biomarker which has not been strictly validated yet [2]. The efficacy of enrichment trials and stratified randomization trials with a time to event variable as the primary endpoint has been compared assuming that the treatment effect reverses between biomarker positive and negative groups and considering subset analysis within each biomarker status group [3].

Phase II trials are to screen out inefficacious treatments before proceeding to a large-scale studies, such as a phase III trial. As such, phase II trials should be completed in a short time period, so that we must choose a small sample size and a short-term surrogate endpoint, such as tumor response or progression-free survival, as the primary endpoint, rather than a confirmatory endpoint, such as overall survival.

In this paper, we demonstrate two phase II cancer clinical trials, one with a predictive biomarker and the other with a prognostic biomarker, and present analysis and sample size calculation methods for these trials. We use a survival variable as the primary endpoint in this paper, but the same concept can be used for any kind of variables including a binary variable, such as tumor response. For the purpose of sample size calculation, we assume exponential survival distributions which are most popularly used in real trial designs, although the statistical testing does not depend on any specific survival distribution. This is a review article of a biostatistical paper [4] with some modifications.

## 2. Materials and Methods

We consider a time to event (or survival) endpoint, progression-free survival (PFS). We use a generalized log-rank test for a trial with an imaging prognostic biomarker and a Cox proportional hazards model for a trial with a predictive biomarker, and derive their sample size formulas. To account for relatively small sample sizes of phase II trials, exact statistical methods are used for binary outcomes, but in general no exact methods are available for survival analysis. Therefore, using simulations on two real trial examples, we evaluate the small sample performance of the discussed statistical tests and their sample size formulas that are derived based on large sample approximation.

## 3. Results

### 3.1. A Phase II Trial with a Predictive Biomarker

Predictive biomarkers help provide information on the likelihood of response to a specific chemotherapy. For example, tumors expressing high thymidylate synthase (TS) levels were shown to be resistant to pemetrexed in a preclinical study [5], but it was not validated by a clinical study yet. Suppose that we want to investigate whether TS expression is a predictive marker for the clinical outcome of pemetrexed/cisplatin (PC) in patients with nonsquamous non–small-cell lung cancer (NSCLC) through a phase II trial. The control non-targeted treatment is gemcitabine/cisplatin (GC). Compared to GC, PC is expected to be similarly efficacious for TS-positive group, but to be more efficacious in TS-negative group.

To investigate this hypothesis, we want to randomize patients between the two treatment arms stratifying by TS-positivity vs. TS-negativity. This trial was designed and published with overall response as the primary endpoint [6], but in this paper, we demonstrate how to design and analyze a trial using PFS as the primary endpoint based on the estimates from the trial.

#### 3.1.1. Statistical Testing

When this study is completed, PFS will be regressed on treatment allocation $z_{1}$ (=0 for GC arm; =1 for PC arm) and TS-positivity $z_{2}$ (=0 for TS-negative group; =1 for TS-positive group) using a proportional hazards model [7]
(1)$$\lambda \left(t\right)=\lambda_{0}\left(t\right)\exp\left(\beta_{1}z_{1}+\beta_{2}z_{2}+\beta_{3}z_{1}z_{2}\right)\;$$

Please note that we have an interaction term $z_{1}z_{2}$ in the model.

From model (1), the hazard functions of four patient groups defined by treatments and TS status are given as $\lambda \left(t|z_{1}=0,z_{2}=0\right)=\lambda_{0}\left(t\right)$, $\lambda \left(t|z_{1}=1,z_{2}=0\right)=\lambda_{0}\left(t\right)\exp\left(\beta_{1}\right)$, $\lambda \left(t|z_{1}=0,z_{2}=1\right)=\lambda_{0}\left(t\right)\exp\left(\beta_{2}\right)$, and $\lambda \left(t|z_{1}=1,z_{2}=1\right)=\lambda_{0}\left(t\right)\exp\left(\beta_{1}+\beta_{2}+\beta_{3}\right)$. For TS-positive patients $\left(z_{2}=1\right)$, the hazard ratio between PC and GC is $\lambda \left(t|z_{1}=1,z_{2}=1\right)/\lambda \left(t|z_{1}=0,z_{2}=1\right)=\exp\left(\beta_{1}+\beta_{3}\right)$, so that we expect $\beta_{1}\approx -\beta_{3}$ if GC and PC are similarly efficacious for TS-positive patients. For GC arm ($z_{1}=0$), the hazard ratio between TS-positivity group and TS-negativity group is $\lambda \left(t|z_{1}=0,z_{2}=1\right)/\lambda \left(t|z_{1}=0,z_{2}=0\right)=\exp\left(\beta_{2}\right)$, so that we will have $\beta_{2}=0$ if GC is non-targeted against TS. With $\beta_{2}=0$, $\lambda_{0}\left(t\right)$ is the hazard function for GC arm. On the other hand, for PC arm ($z_{1}=1$), the hazard ratio between TS-positivity group and TS-negativity group is $\lambda \left(t|z_{1}=1,z_{2}=1\right)/\lambda \left(t|z_{1}=1,z_{2}=0\right)=\exp\left(\beta_{3}\right)$ since GC is non-targeted treatment (i.e., $\beta_{2}=0$). If TS-positive tumors are resistant to pemetrexed, we will have $\beta_{3}>0$. Therefore, the hypotheses of interest are $H_{0}:\beta_{3}=0$ and $H_{1}:\beta_{3}>0$.

For patient i(=1,…,n), let $X_{i}$ be the minimum of censoring time and survival time, $\delta_{i}$ be the event indicator taking 1 if tumor progression has occurred and 0 otherwise, and $Z_{i}={\left(z_{1i},z_{2i},z_{1i}z_{2i}\right)}^{T}$ be the covariate vector. Partial score and information functions for regression coefficients $\beta ={\left(\beta_{1},\beta_{2},\beta_{3}\right)}^{T}$ are given as
$$U\left(\beta \right)=\sum_{i=1}^{n}\int_{0}^{\infty }\left\{Z_{i}-\frac{\sum_{j=1}^{n}Y_{j}\left(t\right)Z_{j}e^{\beta^{T}Z_{j}}}{\sum_{j=1}^{n}Y_{j}\left(t\right)e^{\beta^{T}Z_{j}}}\right\}dN_{i}\left(t\right)$$
and
$$I\left(\beta \right)=\sum_{i=1}^{n}\int_{0}^{\infty }\left[\frac{\sum_{j=1}^{n}Y_{j}\left(t\right)Z_{j}^{⊗2}e^{\beta^{T}Z_{j}}}{\sum_{j=1}^{n}Y_{j}\left(t\right)e^{\beta^{T}Z_{j}}}-\frac{{\left\{\sum_{j=1}^{n}Y_{j}\left(t\right)Z_{j}e^{\beta^{T}Z_{j}}\right\}}^{⊗2}}{{\left\{\sum_{j=1}^{n}Y_{j}\left(t\right)e^{\beta^{T}Z_{j}}\right\}}^{2}}\right]dN_{i}\left(t\right),$$
respectively, where $N_{i}\left(t\right)=\delta_{i}I\left(X_{i}\leq t\right)$ is the event process, $Y_{i}\left(t\right)=I\left(X_{i}\geq t\right)$ is the at-risk process, I(·) is an indicator function, and $z^{⊗2}=zz^{T}$ for a vector *z*.

Let $\hat{\beta }={\left({\hat{\beta }}_{1},{\hat{\beta }}_{2},{\hat{\beta }}_{3}\right)}^{T}$ denote the solution to U(β)=0. Then, $\hat{\beta }$ is approximately normal with mean 0, and variance–covariance $I^{-1}\left(0\right)$ under the global null hypothesis of β=0 [8]. Hence, with a one-sided type I error rate of α, we reject $H_{0}:\beta_{3}=0$ in favor of $H_{1}:\beta_{3}>0$ if ${\hat{\beta }}_{3}/{\hat{\sigma }}_{3}>z_{1-\alpha }$, where ${\hat{\sigma }}_{3}^{2}$ is the (3,3)-component of $I^{-1}\left(0\right)$ and $z_{1-\alpha }$ is the 1−α quantile of the standard normal distribution.

#### 3.1.2. Sample Size Calculation

For sample size calculation of this study, we need to specify following design parameters.

Type I error rate and power, (α,1−β)Allocation proportion for GC arm, $p_{0}$, and for PC arm, $p_{1}$$\left(p_{0}+p_{1}=1\right)$TS-negativity $q_{0}$ and TS-positivity $q_{1}$ based on the prevalence in the study population $\left(q_{0}+q_{1}=1\right)$Assuming exponential distributions for PFS, the hazard rates, $\lambda_{z_{1}z_{2}}$, of the four patient groups, $\lambda_{00}$, $\lambda_{01}$, $\lambda_{10}$, and $\lambda_{11}$Accrual period *a* (or accrual rate *r*) and additional follow-up period *b*

Assuming exponential distributions for PFS with hazard rates $\lambda_{z_{1}z_{2}}$, model (1) is simplified to
$$\lambda =\lambda_{0}\exp\left(\beta_{1}z_{1}+\beta_{2}z_{2}+\beta_{3}z_{1}z_{2}\right)$$
with
$$\lambda_{00}=\lambda_{0},\;\lambda_{10}=\lambda_{0}\exp\left(\beta_{1}\right),\;\lambda_{01}=\lambda_{0}\exp\left(\beta_{2}\right),\;\lambda_{11}=\lambda_{0}\exp\left(\beta_{1}+\beta_{2}+\beta_{3}\right).$$By solving these equations with respect to $\left(\lambda_{0},\beta_{1},\beta_{2},\beta_{3}\right)$, we have
$$\lambda_{0}=\lambda_{00},\;\beta_{1}=\log\lambda_{10}-\log\lambda_{00},\;\beta_{2}=\log\lambda_{01}-\log\lambda_{00},\;$$
(2)$$\beta_{3}=\log\lambda_{11}-\log\lambda_{10}-\log\lambda_{01}+\log\lambda_{00}.\;$$

Hence, we can calculate the values of $\beta_{3}$ under $H_{1}$ in terms of the hazard ratios that are specified as design parameters above.

To derive a sample size formula, we need to calculate the limit of ${\hat{\sigma }}_{3}^{2}$ or $I^{-1}\left(0\right)$ as n→∞ in terms of the design parameters. Let $p_{kl}=P\left(z_{1}=k,z_{2}=l\right)$ for k,l=0 or 1 denote the relative frequency of each cell of the 2×2 table defined by treatment and TS status. Under the stratified randomization scheme, we have $p_{kl}=p_{k}q_{l}$ for k,l=0 or 1.

Appendix A shows that I(0) converges to DA, where D=nd denotes the expected number of events (or number of patients with tumor progression), $d=p_{00}d_{00}+p_{10}d_{10}+p_{01}d_{01}+p_{11}d_{11}$ denotes the probability that a patient has a progression during the study period, $d_{kl}=1-\exp\left(-\lambda_{kl}b\right)\left\{1-\exp\left(-\lambda_{kl}a\right)\right\}$ denotes the probability that a patient in group $\left(z_{1},z_{2}\right)=\left(k,l\right)$ has a progression for k,l=0,1 as derived based on an exponential PFS distribution and U(b,a+b) censoring distribution, and
$$A=\left(\begin{array}{ccc} p_{0}p_{1} & p_{11}-p_{1}q_{1} & p_{0}p_{11}\\ p_{11}-p_{1}q_{1} & q_{0}q_{1} & q_{0}p_{11}\\ p_{0}p_{11} & q_{0}p_{11} & p_{11}\left(1-p_{11}\right) \end{array}\right)$$

Please note that $d_{z_{1}z_{2}}$ is derived from an exponential PFS distribution with hazard ratio $\lambda_{z_{1}z_{2}}$ and a censoring distribution of U(b,a+b). Hence, the limit of ${\hat{\sigma }}_{3}^{2}$ is $\sigma_{3}^{2}=A_{\left(3,3\right)}/D$, where $A_{\left(3,3\right)}$ is the (3,3) component of $A^{-1}$.

From (2), ${\bar{\beta }}_{3}=\log\lambda_{11}-\log\lambda_{10}-\log\lambda_{01}+\log\lambda_{00}$ is the $\beta_{3}$ value specified under $H_{1}$. Since $\left({\hat{\beta }}_{3}-{\bar{\beta }}_{3}\right)/\sigma_{3}$ has the standard normal distribution under $H_{1}$, the power for a local alternative hypothesis $H_{1}:\beta_{3}={\bar{\beta }}_{3}$ is given as
(3)$$1-\beta =P({\hat{\beta }}_{3}/{\hat{\sigma }}_{3}>z_{1-\alpha }|\beta_{3}={\bar{\beta }}_{3})\;=P(\frac{{\hat{\beta }}_{3}-{\bar{\beta }}_{3}}{\sigma_{3}}>z_{1-\alpha }-\frac{{\bar{\beta }}_{3}}{\sigma_{3}}|\beta_{3}={\bar{\beta }}_{3})\;=\bar{\Phi }\left(z_{1-\alpha }-{\bar{\beta }}_{3}/\sigma_{3}\right)\;$$
where $\bar{\Phi }\left(\cdot \right)$ is the survivor function of the standard normal distribution.

Noting that $\sigma_{3}^{2}=DA_{\left(3,3\right)}=ndA_{\left(3,3\right)}$, we obtain the required number of events
$$D=A_{\left(3,3\right)}{\left(\frac{z_{1-\alpha }+z_{1-\beta }}{{\bar{\beta }}_{3}}\right)}^{2}$$
or the required sample size
(4)$$n=\frac{A_{\left(3,3\right)}}{d}{\left(\frac{z_{1-\alpha }+z_{1-\beta }}{{\bar{\beta }}_{3}}\right)}^{2}\;$$
by solving Equation (Equation 3).

Formula (4) requires specification of accrual period *a* together with (α,1−β), ${\bar{\beta }}_{3}$, *b*, $p_{0}$ and $q_{0}$. In designing a clinical trial, however, we can estimate the accrual pattern, rather than an accrual period. Suppose that patients are expected to be enrolled to the study at a rate of *r* during an accrual period based on the number of patients treated by the study member sites recently. Assuming uniform patient accrual during period *a*, we have n≈a×r. Noting that d=d(a) is a function of *a*, (4) is expressed as
(5)$$a\times r=\frac{A_{\left(3,3\right)}}{d(a)}{\left(\frac{z_{1-\alpha }+z_{1-\beta }}{{\bar{\beta }}_{3}}\right)}^{2}\;$$

By solving (5) with respect to *a* using a numerical method, such as the bisection method, we obtain the required accrual period, say $a^{*}$, and the required sample size $n=a^{*}r$.

#### 3.1.3. Example 1

We demonstrate our sample size calculation method with the NSCLC trial that is introduced above. We will randomize patients between the two treatment arms with 1-to-1 fashion, i.e., $p_{0}=p_{1}=1/2$ stratified by TS status. The expected TS-positivity is 50% (i.e., $q_{0}=q_{1}=0.5$) because the median TS level was selected as the cutoff value for TS-positivity from a previous study [9]. Hence, we have $p_{z_{1}z_{2}}=p_{z_{1}}q_{z_{2}}=1/4$ for $z_{1},z_{2}=0$ or 1. The 6-month PFS is expected to be about 35% for GC arm regardless of TS level and for PC arm with TS-positivity, and 55% for PC arm with TS-negativity. For an exponential distribution, *t*-year survival probability S(t) is associated with its hazard rate λ by S(t)=exp(−λt). Therefore, the annual hazard rates under the alternative hypothesis are given as $\lambda_{00}=\lambda_{01}=\lambda_{11}=2.100$ and $\lambda_{10}=1.196$ under the exponential PFS assumption. For these hazard rates, we have the baseline hazard rate $\lambda_{0}=2.100$, $\beta_{2}=0$, and $\beta_{3}=-\beta_{1}=0.563$ from (2). Suppose that about 10 patients per month are expected to be entered to the study, i.e., an annual accrual rate of r=120. We plan to follow the patients for additional b=1 year after the last patient enters. Then, the 1-sided α=0.1 test for $H_{0}:\beta_{3}=0$ against $H_{1}:\beta_{3}=0.563$ in model (1) requires n=345 patients for a power of 1−β=0.9. The expected number of events (i.e., number of patients with a disease progression) at the analysis will be D=333. As an effort to lower the sample size for this phase II trial, we use a large α level compared to the standard two-sided α=0.05. We observe an empirical power of 0.897 from 10,000 simulation samples of size n=345 that are generated at the design setting. This trial recruited 321 patients using overall response as the primary endpoint [6].

A stratified randomized trial of a treatment with a predictive biomarker requires a large sample size for testing on the interaction term. Sample size of a trial for testing the interaction term with 50% of biomarker positivity may be compared to that of a trial for an arm-to-arm comparison with 1-to-3 randomization in the setting of the NSCLC trial expecting a higher efficacy of PC only for TS-negative group. Let us consider a randomized trial to compare two treatment arms with (α,1−β,r,b)=(0.1,0.9,120,1) as above and 6-month PFS of 35% for the control treatment and 55% for the experimental treatment. In this case, we need only n=122 (D=102) patients by 1-to-3 randomization by a sample size formula for the standard log-rank test [10]. If TS had been already validated to be a predictive biomarker of pemetrexed before this trial, then we could have chosen an enrichment trial for TS-negative patients that would require a much smaller sample size. The efficiency has been compared between an enrichment design and a stratified randomization design has been for predictive biomarker in terms of a continuous outcome [11].

### 3.2. A Phase II Trial with a Prognostic Biomarker

Prognostic biomarkers provide information on the overall cancer outcome in patients to facilitate cancer diagnosis regardless of selected treatments. In this section, we consider a phase II trial with an imaging prognostic biomarker. Chemotherapy B has been a standard regimen for patients with non-bulky stage I and II Hodgkin lymphoma. In a previous study on 6 cycles of B, each patient had a FDG-PET (fluorodeoxyglucose positron-emission tomography) imaging after 2 cycles of B. It was found that the patients with a negative PET image (group 1) and those with a positive PET image (group 2) had a 3-year PFS of $S_{1}\left(3\right)=0.86$ and $S_{2}\left(3\right)=0.52$, respectively, and the hazard ratio, $\Delta =\lambda_{2}/\lambda_{1}$, was estimated as $\Delta_{0}=4.3$.

In a new single-arm phase II trial, the patients with a negative PET image after 2 cycles of B will be treated by additional 4 cycles of the chemotherapy B as in the previous study, whereas those with a positive PET image after 2 cycles of B will be treated by 4 cycles of a more aggressive chemotherapy C plus radiation therapy (C+RT).

In this trial, we want to show that by treating PET positive patients with the more aggressive therapy C+RT, their PFS will become closer to that of PET negative patients who are treated by the standard chemotherapy B. To this end, we test $H_{0}:\Delta =\Delta_{0}$ against $H_{1}:\Delta <\Delta_{0}$. Although the PFS of group 2 will be different between $H_{0}$ and $H_{1}$, that of group 1 is expected to be identical since PET negative patients receive the same treatment as that of the previous study.

#### Statistical Testing

Let $n_{k}$ denote the sample size in group *k*, $n=n_{1}+n_{2}$ the total sample size, and $T_{ki}$ the time to progression for subject *i* in group *k* ($1\leq i\leq n_{k};k=1,2$). We observe $\left(X_{ki},\delta_{ki}\right)$, where $X_{ki}$ is the minimum of $T_{ki}$ and the censoring time and $\delta_{ki}$ is an event (or progression) indicator taking 1 if the subject had a tumor progression and 0 otherwise. For group *k*, $T_{k1},\ldots ,T_{k,n_{k}}$ are distributed with hazard function $\lambda_{k}\left(t\right)$. Under the proportional hazards assumption, $\Delta =\lambda_{2}\left(t\right)/\lambda_{1}\left(t\right)$ denotes the hazard ratio between the two patient groups.

Let ${\hat{\Lambda }}_{k}\left(t\right)=\int_{0}^{t}Y_{k}^{-1}\left(t\right)dN_{k}\left(t\right)$ denote the Aalen–Nelson estimator [12,13] for the cumulative hazard function $\Lambda_{k}\left(t\right)=\int_{0}^{t}\lambda_{k}\left(s\right)ds$, $Y_{k}\left(t\right)=\sum_{i=1}^{n_{k}}I\left(X_{ki}\geq t\right)$ and $N_{k}\left(t\right)=\sum_{i=1}^{n_{k}}\delta_{ki}I\left(X_{ki}\leq t\right)$ are the at-risk process and the event process for group *k*, respectively, and $N\left(t\right)=N_{1}\left(t\right)+N_{2}\left(t\right)$. It was shown [14] that
$$W\left(\Delta \right)=\int_{0}^{\infty }\frac{Y_{1}\left(t\right)Y_{2}\left(t\right)}{Y_{1}\left(t\right)+\Delta Y_{2}\left(t\right)}\left\{\Delta d{\hat{\Lambda }}_{1}\left(t\right)-d{\hat{\Lambda }}_{2}\left(t\right)\right\}$$
is increasing in Δ, and $W\left(\Delta \right)/\sigma_{n}\left(\Delta \right)$ is asymptotically N(0,1), where
$$\sigma_{n}^{2}\left(\Delta \right)=\Delta \int_{0}^{\infty }\frac{Y_{1}\left(t\right)Y_{2}\left(t\right)}{{\left\{Y_{1}\left(t\right)+\Delta Y_{2}\left(t\right)\right\}}^{2}}dN\left(t\right)$$

Hence, we reject $H_{0}:\Delta =\Delta_{0}$, in favor of $H_{1}:\Delta <\Delta_{0}$, if $W\left(\Delta_{0}\right)/\sigma_{n}\left(\Delta_{0}\right)>z_{1-\alpha }$ with one-sided type I error rate α. The test statistic with $\Delta_{0}=1$ is the standard log-rank test [15].

### 3.3. Sample Size Calculation

We want to estimate the sample size *n* under a local alternative hypothesis $H_{1}:\Delta =\Delta_{1}$ (<$\Delta_{0})$ with a desired power. For sample size calculation of this trial, we need to specify following design parameters.

Type I error rate and power, (α,1−β)PET-negativity and PET-positivity, $p_{1},p_{2}$Distributions of PFS for PET negative and PET positive patient groups: exponential distributions with hazard rates $\lambda_{1}$ for PET negative group; $\lambda_{20}$ under $H_{0}$ and $\lambda_{21}$ under $H_{1}$ for PET positive groupAccrual period *a* (or accrual rate *r*) and additional follow-up period *b*

Using the specified hazard rates, we have $\Delta_{0}=\lambda_{20}/\lambda_{1}$ and $\Delta_{1}=\lambda_{21}/\lambda_{1}$. A sample size formula with $\Delta_{0}>1$ and $\Delta_{1}=1$ for designing non-inferiority trials was proposed [14]. This sample size formula was further extended for general $\Delta_{0}$ and $\Delta_{1}$ with $\Delta_{1}<\Delta_{0}$ [16].

Appendix B derives a sample size formula by adapting Jung and Chow’s formula [16] for this trial,
(6)$$n=\frac{{\left(\sigma_{0}z_{1-\alpha }+\sigma_{1}z_{1-\beta }\right)}^{2}}{\omega^{2}}\;$$
where
$$\omega =p_{1}p_{2}\int_{0}^{\infty }\frac{G\left(t\right)S_{1}\left(t\right)S_{21}\left(t\right)\left\{\lambda_{1}\Delta_{0}-\lambda_{21}\right\}}{p_{1}S_{1}\left(t\right)+p_{2}\Delta_{0}S_{21}\left(t\right)}dt$$
$$\sigma_{0}^{2}=\Delta_{0}p_{1}p_{2}\int_{0}^{\infty }\frac{G\left(t\right)S_{1}\left(t\right)S_{21}\left(t\right)\left\{p_{1}\lambda_{1}S_{1}\left(t\right)+p_{2}\lambda_{21}S_{21}\left(t\right)\right\}}{{\left\{p_{1}S_{1}\left(t\right)+p_{2}\Delta_{0}S_{21}\left(t\right)\right\}}^{2}}dt$$
$$\sigma_{1}^{2}=p_{1}p_{2}\int_{0}^{\infty }\frac{G\left(t\right)S_{1}\left(t\right)S_{21}\left(t\right)\left\{p_{2}\lambda_{1}\Delta_{0}^{2}S_{21}\left(t\right)+p_{1}\lambda_{21}S_{1}\left(t\right)\right\}}{{\left\{p_{1}S_{1}\left(t\right)+p_{2}\Delta_{0}S_{21}\left(t\right)\right\}}^{2}}dt$$$S_{1}\left(t\right)=\exp\left(-\lambda_{1}t\right)$, $S_{21}\left(t\right)=\exp\left(-\lambda_{21}t\right)$, G(t) is the survivor function of the U(b,a+b) censoring distribution with
$$G\left(t\right)=\left\{\begin{array}{cc} 1 & \mathrm{if}\;t\leq \;b\\ -t/b+(a+b)/a & \mathrm{if}\;b<t\leq a+b\\ 0 & \mathrm{if}\;t>a+b \end{array}\right.$$
and $p_{k}=n_{k}/n$. The integrals for ω, $\sigma_{0}^{2}$, and $\sigma_{1}^{2}$ are calculated using a numerical method.

The number of events *D* expected at the analysis time under $H_{1}$ is calculated by $D=n(p_{1}d_{1}+p_{2}d_{2})$, where $d_{1}=1+\int_{0}^{\infty }S_{1}\left(t\right)dG\left(t\right)=1-\exp\left(-\lambda_{1}b\right)\left\{1-\exp\left(-\lambda_{1}a\right)\right\}$ and $d_{2}=1+\int_{0}^{\infty }S_{21}\left(t\right)dG\left(t\right)=1-\exp\left(-\lambda_{21}b\right)\left\{1-\exp\left(-\lambda_{21}a\right)\right\}$.

Sample size formula (6) assumes that the accrual period *a* is specified. Suppose that accrual rate *r* is specified instead of accrual period *a*. Given $\left(\lambda_{1},\Delta_{0},\Delta_{1},\alpha ,1-\beta ,p_{1},b\right)$, ω=ω(a) and $\sigma_{h}=\sigma_{h}\left(a\right)$ for h=0,1 are functions of *a*. Under uniform accrual assumption, we have n=a×r. Hence, (6) is expressed as
(7)$$a\times r=\frac{{\left\{\sigma_{0}\left(a\right)z_{1-\alpha }+\sigma_{1}\left(a\right)z_{1-\beta }\right\}}^{2}}{\omega^{2}\left(a\right)}.\;$$

By solving (7) with respect to *a*, using a numerical method such as the bisection method, we obtain the required accrual period $a^{*}$ and the required sample size $n=a^{*}\times r$.

#### Example 2

We consider the PET-guided Hodgkin lymphoma trial introduced in the beginning of this section. Under $H_{0}$, we assume a 3-year PFS of 86% and 52% for PET negative and positive groups, respectively, which correspond to annual hazard rates of $\left(\lambda_{1},\lambda_{20}\right)=\left(0.050,0.218\right)$ under an exponential PFS model, resulting in a hazard ratio of $\Delta_{0}=4.3$. By treating the PET positive patients with an aggressive treatment C+RT, we expect to increase their 3-year PFS up to 74% (from 52%), resulting in an annual hazard rate of $\lambda_{21}=0.100$ and hazard rate of $\Delta_{1}=2.0$. The previous study observed about $p_{2}=20\%$ of PET-positivity. Assuming an annual accrual rate of r=60 patients and b=3 years of additional follow-up after completion of accrual, we need n=191 patients for 1−β=90% power for detecting $H_{1}:\Delta_{1}=2$ by the generalized log-rank test $W\left(\Delta_{0}\right)/\sigma_{n}\left(\Delta_{0}\right)$ with one-sided α=10% under $H_{0}:\Delta_{0}=4.3$. Under this specific alternative hypothesis, we expect about 46 events (progressions or deaths) at the data analysis. This trial was conducted with this study objective as a second objective [17]. Simulation studies are conducted to evaluate the performance of the calculated sample size under the above design settings under $H_{0}$ and $H_{1}$. Using 10,000 simulation samples of size n=191 under each hypothesis, the empirical type I error rate and power are observed as 0.0984 (to be compared to α=0.1) and 0.8749 (to be compared to 1−β=0.9), respectively.

## 4. Discussion

We have presented design and analysis methods of two phase II trials for biomarker-guided treatments.

The power of the statistical tests of the trials discussed above depends on the prevalence of the biomarker positivity. Therefore, we need to check the observed prevalence during the patient accrual, and to recalculate the sample size if the observed prevalence is very different from the one specified at the design stage. For both of our example trials, the initial sample size will be under-powered if the observed prevalence is farther from 1/2 than the specified one at the design stage. In this case, we may plan a sample size recalculation reflecting the observed prevalence in the middle of the trial, and modify the sample size of the trial if necessary.

For the sample size calculations, we have assumed exponential survival distributions and an accrual pattern with a constant accrual rate, but we can easily extend the formulas for any survival distributions and any accrual pattern [18]. We have considered a survival endpoint as the primary endpoint in this paper, but the concept can be used to design and analysis for biomarker-driven phase II trials with other type of endpoint, such as a binary outcome for tumor response.

As an effort to lower the sample size of a phase II trial from that of a phase III trial, we use a high type I error rate [19,20], such as 1-sided α=5% or 10% (compared to 2-sided α=5%), a surrogate short-term outcome, such as tumor response or progression-free survival (compared to a confirmatory endpoint such as overall survival), a larger treatment effect, and a single-arm design (compared to a randomized design). Despite these efforts, we observe that a randomized trial stratified by a predictive biomarker requires a relatively large sample size for a phase II trial. This fact is pointed out by literature [3,11,21].

## 5. Conclusions

Through simulations on the two real study examples, we find that the proposed statistical tests control the type I error rate accurately and the calculated sample sizes maintain the appropriate power. The sample size calculations require some numerical methods for integration and solving equations. The author developed Fortran programs to implement the sample size formulas, which are available upon request.

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
