# Peer review of "Design and Analysis of Cancer Clinical Trials for Personalized Medicine"

_jpm, 2021, doi:10.3390/jpm11050376_

Round 1

Reviewer 1 Report

In general, the revised manuscript is improved, and the changes enhanced the quality of the manuscript. All other comments were well addressed except for point 2. The author pointed out in his response to the previous comment (Point 2) that solving simple equation using the bisection method may not have limitations or drawbacks. As pointed out by the author, the use of numerical method to solve a simple equation used in his investigation was straightforward, it will be worthwhile that the author adds to include the statement into the manuscript to bring out this strength for using numerical method in his study. This would also rationalize to why numerical methods were applied to evaluate the two real clinical trials relating to personalized medicine.

Author Response

In order to address this comment, the last sentence of Conclusions section is slightly changed to:

“The sample size calculations require some numerical methods for integration and solving equations. The author developed Fortran programs to implement the sample size formulas, which are available upon request.”

Reviewer 2 Report

This manuscript addresses the design and analysis of statistical methods for biomarker-guided treatments. The author has extensively explained the two different types of the statistical approach in the manuscript.

The manuscript should be edited for few omissions. For example, in the materials and method section (repetition of words: For a trial For a trial).

Author Response

Addressing this comment, I have read the manuscript carefully and polished further. All changes (including the one pointed out by Reviewer #2, “for a trial for a trial” to “for a trial”) are colored in red in the revised manuscript.  

This manuscript is a resubmission of an earlier submission. The following is a list of the peer review reports and author responses from that submission.

Round 1

Reviewer 1 Report

The manuscript presented the design and analyses (i.e., statistical testing methods and sample size calculation methods) of two phase II cancer clinical trials for biomarker-guided treatments. I have the following suggestions:

  1. In the summary section, the author can include a statement to convey the core findings of this study. The author can expand on the approach of the statistical testing methodology and sample size calculation methodology used for designing and analysing phase II cancer clinical trials with predictive biomarker and with prognostic biomarker. How would the design and analyses between phase II cancer clinical trials with predictive biomarker differ from those with prognostic biomarker? Providing a detailed summary may help to rationalize a more suitable approach to take when designing a phase II cancer clinical trial study. The author should also discuss the differences in the methodology with phase II cancer clinical trials with predictive biomarker and with prognostic biomarker in the discussion.
  2. The author stated in section 3.3 that the use of numerical method such as bisection method was applied for calculating the required accrual period and the required sample size. It is suggested that the author can specify clearly the different numerical methods used in the written manuscript. The author can further discuss on the limitation/drawbacks of the numerical method used for the analyses and provide suggestion to minimize errors and approximations in the numerical methods.
  3. In the discussion section, the author indicated that to prevent under-powered statistics, one could plan to increase the sample size of their study by up to 30% depending on the sample size recalculation. Why did the author suggest for up to 30% increment in the sample size? Does 30% increment in the sample size result in an over-powered study? Can the author provide references to this statement?
  4. Minor comment: Lowercase ‘w’ for ‘we’ in the first line of the conclusion section.

Reviewer 2 Report

I am afraid this review topic is beyond the remit of this journal. Although article it self has a very important thesis and indeed present an important basis for personalising treatments, in the current form it is vague and does not fit the remit of this journal. First example of biomarker used in the article is an acceptable one. However, second example used in article is not an acceptable one and does not match their hypothesis. I suggest that this article should be submitted to journal focused on biomarkers.